# Human–Robot Interaction Using Learning from Demonstrations and a Wearable Glove with Multiple Sensors

**DOI:** 10.3390/s23249780

**Published:** 2023-12-12

**Authors:** Rajmeet Singh, Saeed Mozaffari, Masoud Akhshik, Mohammed Jalal Ahamed, Simon Rondeau-Gagné, Shahpour Alirezaee

**Affiliations:** 1Mechanical, Automotive, and Material Engineering Department, University of Windsor, Windsor, ON N9B 3P4, Canada; rsbhourji@uwindsor.ca (R.S.); saeed.mozaffari@uwindsor.ca (S.M.); akhshik@uwindsor.ca (M.A.); m.ahamed@uwindsor.ca (M.J.A.); 2Department of Chemistry and Biochemistry, University of Windsor, Windsor, ON N9B 3P4, Canada; simon.rondeau-gagne@uwindsor.ca

**Keywords:** robotic grasping, human–robot interaction, inertia, pressure, flexi sensors, wearable devices, learning from demonstration

## Abstract

Human–robot interaction is of the utmost importance as it enables seamless collaboration and communication between humans and robots, leading to enhanced productivity and efficiency. It involves gathering data from humans, transmitting the data to a robot for execution, and providing feedback to the human. To perform complex tasks, such as robotic grasping and manipulation, which require both human intelligence and robotic capabilities, effective interaction modes are required. To address this issue, we use a wearable glove to collect relevant data from a human demonstrator for improved human–robot interaction. Accelerometer, pressure, and flexi sensors were embedded in the wearable glove to measure motion and force information for handling objects of different sizes, materials, and conditions. A machine learning algorithm is proposed to recognize grasp orientation and position, based on the multi-sensor fusion method.

## 1. Introduction

Industrial robots have been used in a variety of applications, ranging from manufacturing to healthcare settings. These highly capable machines are designed to perform repetitive tasks with precision and efficiency, enhancing productivity and reducing human labor in industries. While traditionally programmed for specific tasks, the evolving needs of modern industries have led to the demand for collaborative robots (cobots) that can work alongside humans, thus emphasizing the importance of human–robot interaction (HRI) [1]. Collaborative robots are designed to safely and effectively collaborate with human operators, enhancing productivity and flexibility [2]. HRI applications have gained significant attention across various domains. In manufacturing and industrial settings, robots collaborate with humans, assisting in tasks, such as assembly, material handling, and quality control, improving productivity and worker safety [3]. Additionally, robots are employed in healthcare centers, for patient care, rehabilitation, and surgical assistance, enhancing medical procedures and reducing the physical strain on healthcare professionals [4]. In the service industry, robots function as receptionists, guides, or companions, delivering personalized assistance and improving customer experiences [5]. HRI also finds applications in education, entertainment, and social interactions, where robots serve as tutors, performers, or companions, fostering engagement and enriching human experiences [6]. 

Learning from demonstration (LfD) is a special case of HRI, which enables robots to learn new skills and behaviors through human guidance, enhancing the capabilities of industrial robots, enabling them to interact with humans and optimizing their performance [7]. The rapid development of sensor technologies and integration of artificial intelligence and machine learning techniques have opened up new possibilities in LfD [8]. Demonstration techniques can be divided into the following main groups [9]: kinesthetic teaching, teleoperation demonstration, learning from observations, and sensor-based methods. 

The kinesthetic teaching method involves physical guidance, where a human physically moves the robot’s limbs or manipulates its body to demonstrate the desired behavior. The robot learns by recording and reproducing the demonstrated movements. Since the human teacher directly guides the robot, the kinematic boundaries of the robot, including joint limits and workspace, are considered during the demonstration. In other words, the real-time feedback provided by the human teacher during kinesthetic teaching allows for immediate corrections and refinements to the robot’s movements, facilitating faster learning and performance improvements. However, kinesthetic teaching requires a significant dependency on human guidance, which makes it time-consuming and labor-intensive. The effect of the human teacher’s proficiency on their demonstration data and kinesthetic teaching was investigated in [10]. Scaling up kinesthetic teaching to complex or multi-step tasks can also become challenging, as the complexity increases, and it becomes more difficult for a novice human teacher to provide precise physical guidance.

In teleoperation demonstrations, robots are controlled remotely by the human operator via a control interface, such as a joystick. The operator’s inputs are transmitted to the robot in real-time, enabling the robot to mimic the operator’s movements and actions. Unlike kinesthetic teaching, in which the human teacher moves individual segments by hand, teleoperation demonstration requires an input device for robot movement. The impact of the input device on the performance of teleoperation was studied in [11]. On the negative side, teleoperation may not be suitable for tasks that require high levels of autonomy or decision making, as the robot relies heavily on the operator’s guidance. It is shown that teleoperation is slower and more challenging in comparison with kinesthetic teaching [11]. An overview of teleoperation techniques with a focus on skill learning and their application to complex manipulation tasks is presented in [12]. Key technologies, such as manipulation skill learning, multimodal interfacing for teleoperation, and telerobotic control, are discussed. 

Unlike other learning methods that rely on explicit demonstrations or interactions, learning from observations leverages existing data to extract patterns, make inferences, and acquire knowledge. In this approach, the robot typically receives a set of observations in the form of images or videos. Then, machine learning algorithms are employed to analyze the data and extract meaningful patterns or relationships. In [8], an LfD system based on computer vision was developed to observe human actions and deep learning to perceive the demonstrator’s actions and manipulated objects. Deep neural networks were utilized for object detection, object classification, pose estimation, and action recognition. The proposed vision-based LfD method was tested on an industrial robot, Fanuc LR Mate 200ic, to pick and place objects with different shapes. The experimental results showed 95% accuracy. The demonstrator’s intention was recognized by analyzing 3D human skeletons captured by RGB-D videos [13]. It was shown that modeling the conditional probability of interaction between humans and robots in different environments can lead to faster and more accurate HRI. However, the presence of occlusion poses several problems for vision-based HRI systems. It can lead to incomplete or ambiguous visual information, making it challenging to accurately track and recognize objects or individuals.

Although the above demonstration techniques have greatly facilitated the interaction between humans and robots, they cannot completely meet the demands of HRI in dynamic and complex environments. Recently, wearable sensing technology has been applied in HRI. Wearable sensing technology is an example of sensor-based methods, which rely on various kinds of sensors to perceive users’ status [14]. Wearable devices are usually equipped with inertial, bending, electromyography (EMG), and tactile sensors to capture position, orientation, motion, and pressure. A data glove is a type of wearable device commonly utilized in HRI to capture and transmit real-time action postures of the human hand to the robot. The most employed approach in detecting finger bending involves the integration of flex sensors that are attached to the finger joint positions on the hand [15]. A wearable glove system based on 12 nine-axis inertial and magnetic unit (IMMU) sensors was proposed for hand function assessment [16]. The sensory system captured the acceleration, angular velocity, and geomagnetic orientation of human hand movements. In [17], a wearable data glove was designed to sense human handover intention information based on fuzzy rules. The glove contains six six-axis IMUs, one located on the back of the hand and the rest on the second section of each finger.

This paper proposes an experimental HRI system to grab different objects with a robotic hand. Human demonstrators grasp objects with a wearable glove equipped with an accelerometer, flex sensor, and pressure sensor. The data glove provides useful information to train machine learning algorithms to mimic human grasping actions. In addition, an HRI module is evaluated by a universal robot. To evaluate the effectiveness of this sensor-based HRI system, different objects with different shapes were grasped by the robotic hand. 

## 2. Methodology 

In this section, we will provide a detailed description of the proposed data glove, encompassing its hardware and software components, as well as the sensors utilized. Additionally, we will delve into the calibration process of the sensors, the coordinate systems employed, and the placement of the sensors on the glove.

### 2.1. System Hardware and Software

The proposed HRI system consists of MEMS sensors, a microprocessor, a host computer, and a robotic hand. Figure 1 illustrates the communication flow between the hardware. The ADXL335 MEMS (microelectromechanical system) inertial sensor has been selected as the sensing module for capturing finger motion information. It is a compact, slim, low-power accelerometer that provides a fully integrated 3-axis accelerometer with voltage output that is conditioned for signal processing. It features ±3 g tri-axis accelerometer readings. The second sensor is the flex sensor that is employed to detect the bending of the first finger segment. We also used SPX-14687 sensor, which is a pressure sensor working based on the principle of proximity. Table 1 shows the specifications of each sensor. The raw sensor data of multiple ADXL335, flex, and SPX-14687 are sampled at 100 Hz with I2C interface by using low-power processor RP2040. The incorporation of flexible GPIO (General Purpose Input/Output) in our system enables the connection of various digital interfaces, including SPI (Serial Peripheral Interface) Master/slave, TWI (Two-Wire Interface) Master, and UART (Universal Asynchronous Receiver–Transmitter). These GPIOs provide the flexibility to connect and communicate with external devices or modules that utilize these digital interfaces. The host computer receives the data transmitted by the processor through the serial port and subsequently performs the necessary calculations. The robot hand receives action commands to execute the specific applications.

Figure 2a illustrates a system schematic of the data glove, depicting the arrangement of sensors and the communication network. In Figure 2b, the prototype of the data glove is presented, showcasing various versions and the interface with a robotic hand. In the initial version of the data glove, flex sensors were employed to measure finger bending. To capture the bending values of the finger segments, a rigid printed circuit board (PCB) was designed. In the second version, a significant improvement was made by developing a compact and flexible PCB. This flexible PCB design allowed for a more ergonomic and comfortable fit on the hand, while still effectively recording the bending data of the fingers. In the third version of the data glove, additional sensors were incorporated to further enhance the accuracy and precision of data capture, particularly during grasping actions. In addition to the flex sensors, an accelerometer and pressure sensors were integrated into the glove. The accelerometer measured the hand’s orientation and movement in three-dimensional space, providing additional information about hand motion. The pressure sensors detected the pressure or force applied by the fingers during grasping actions, enabling more detailed and comprehensive data capture.

### 2.2. Coordinate System 

In our system design, there are three coordinate systems that interact through mutual transformations: the world frame (w-frame), the hand frame (h-frame), and the sensor frame (s-frame), as depicted in Figure 3. The definitions of these three frames are as follows.

*w-frame* is defined with the Earth as the reference for describing human posture. In this study, we adopt the local Earth frame as the w-frame, where the X-axis aligns with the north direction, the Y-axis aligns with the east direction, and the Z-axis is perpendicular to the ground plane, pointing towards the center of the Earth.*s-frame* represents the coordinate system defined by the manufacturer during the sensor’s design. In our study, s-frame is determined by the ADXL335 inertial sensor. The specific definitions of these coordinate systems are illustrated in Figure 3.*h-frame* is primarily employed to characterize the spatial posture of the hand and individual finger segments. In this paper, we consider the palm as the root node and establish five kinematic chains extending to each finger based on the hand’s skeletal structure (Figure 3). The segments are connected through the joint points, and each joint is associated with its respective frame to represent the spatial posture of the segment.

### 2.3. Sensor Placement

The placement of sensors plays a crucial role in the development of the data glove. Figure 4a depicts the structure of the hand joints. The joints in the human hand’s index finger, middle finger, ring finger, and little finger can be classified into three main categories: distal interphalangeal joint (DIP), proximal interphalangeal joint (PIP), and metacarpophalangeal joint (MP). However, the thumb differs in having only two joints: interphalangeal joint (IP) and metacarpophalangeal joint (MP). In the work by Fei et al. [16], they positioned 12 IMU sensors on the hand phalanges, while an additional IMU sensor was placed on the back of the hand to estimate joint angles and hand gestures. In this research paper, we find the optimum position of sensors by conducting grasping experiment using objects with different shapes and sizes, as shown in Figure 4b–d. Based on the data obtained from these experiments, we proposed placing accelerometer sensors on the second segment of fingers, specifically between the PIP joint and DIP joint. This segment is primarily responsible for strong grasping actions. Therefore, we use 5 accelerometer sensors and flex sensors for capturing grasping actions, reducing the number of sensors compared to the previous work [16]. The flex, accelerometer, and pressure sensor placement for single finger is depicted in Figure 5. 

### 2.4. Sensor Calibration

The accelerometer may encounter several errors, including zero deviation error, scale factor error, and non-orthogonal error. These errors contribute to the overall accelerometer error model, which can be represented as shown in reference [17].
(1)$$\left[\begin{array}{c} a_{x\_c}\\ a_{y\_c}\\ a_{z\_c} \end{array}\right]=\left[\begin{array}{ccc} S_{11} & S_{12} & S_{13}\\ S_{12} & S_{22} & S_{23}\\ S_{13} & S_{23} & S_{33} \end{array}\right]\times \left[\begin{array}{c} a_{x_{\mathrm{raw}}}-b_{x}\\ a_{y_{\mathrm{raw}}}-b_{y}\\ a_{z_{\mathrm{raw}}}-b_{z} \end{array}\right]$$

In Equation (1), $a_{x_{raw}},a_{y_{raw}},a_{z_{raw}}$ show the raw data obtained from the accelerometer and $a_{x\_c},a_{y\_c},a_{z\_c}$ represent the calibrated acceleration data. The terms $b_{x},b_{y},b_{z}$ correspond to the bias correction values, while $S_{ij}\left(i=1,2,3;j=1,2,3\right)$ represents the scale factor and nonorthogonality correction factor values. The calibration error, denoted as $e\left(S_{11},\ldots \ldots \ldots .S_{33},\;b_{x},\ldots \ldots \ldots .\;b_{z}\right)$, is defined as the discrepancy between the squared sum of the calibrated acceleration and the squared gravitational acceleration:(2)$$e\left(S_{11},\ldots \ldots \ldots .S_{33},\;b_{x},\ldots \ldots \ldots .\;b_{z}\right)=a^{2}{}_{x\_c}+a^{2}{}_{y\_c}+a^{2}{}_{z\_c}-g^{2}$$

Gauss–Newton’s method is applied to estimate the unknown calibration parameters $e\left(S_{11},\ldots \ldots \ldots .S_{33},\;b_{x},\ldots \ldots \ldots .\;b_{z}\right)$. The iteration of calibration error can be represented by the following equation: (3)$$e_{k+1}=e_{k}+\gamma_{k}d_{k}$$
where $\gamma_{k}$ is the damping control factor, which is used to control the convergence speed of the iteration algorithm. A larger $\gamma_{k}$ value means a faster convergence speed but lower accuracy. $d_{k}$ is defined as the iterative direction:(4)$$d_{k}=\left(J^{T}J^{-1}\right)\left(J^{T}\left(-e\right)\right)$$
where matrix *J* is defined as the Jacobian matrix of the calibration error: (5)$$J\left(e\right)=\frac{\delta e}{\delta x}=\left[\begin{array}{ccc} \frac{\delta e}{\delta s_{11}} & \ldots & \frac{\delta e}{\delta b_{z}}\\ \vdots & \ddots & \vdots \\ \frac{\delta e}{\delta s_{11}} & \ldots & \frac{\delta e}{\delta b_{z}} \end{array}\right]$$

When the iteration time reaches its maximum or the convergence condition is satisfied, the unknown calibration parameters can be estimated. With ζ defined as the convergence threshold, the convergences condition can be represented as follows: (6)$$\left|e\left(S_{11},\ldots \ldots \ldots .S_{33},\;b_{x},\ldots \ldots \ldots .\;b_{z}\right)\right|<\zeta$$

The Arduino code is utilized to capture the raw data of the accelerometer by moving the ADXL335 sensor along the X-Y-Z directions. Subsequently, a Python code is developed to estimate the bias correction values, as well as the scale factor and nonorthogonality correction factor values. Figure 6 illustrates the comparisons of normalized acceleration data before and after calibration. As can be seen, the accuracy of the local gravitational acceleration is enhanced after calibration.

### 2.5. Experimental Results of Sensor Calibration

To verify the accuracy of the joint angles obtained from the data glove accelerometer (ADXL335), a manual drawing protector is employed to measure the angles. The validation process is depicted in Figure 7. It is worth noting that the angle values are represented as negative due to the utilization of a second quadrant system.

Table 2 presents a comparison of the measurement results obtained from two sources: the drawing protector and the data glove accelerometer. Around 20 samples were collected for 90- and 30-degree positions. The angles recorded correspond to the index finger when held in two distinct positions. It is important to note that only the rotation of the Y-axis of the accelerometer was considered for this specific experiment. The average error rate was less than 2% and the maximum deviation was nearly 1.55 degrees. 

## 3. Human–Robot interaction Using Data Glove

To verify the functionality of the developed data glove, real-time human–robot interaction is performed. In this research paper, a robotic hand is attached to the end of a universal robot (UR5) to replicate the motions performed by the human demonstrator. The robotic hand employed in this study is equipped with five servo motors to control each finger. It is connected to the end of the universal robot (UR5) with 3D printed mounting and operated through the data glove. Figure 8 shows the components in our HRI facilitated by the data glove.

Moreover, this paper proposes a learning-from-demonstration method to develop a machine learning model based on the sensor data from the data glove during various grasping operations. This approach aims to enhance the understanding and capabilities of HRI through machine learning techniques. 

This study concentrates on objects with different shapes and sizes. Figure 9 illustrates a flow diagram of the machine learning model proposed in this paper, which incorporates a learning-from-demonstration approach. The objects are categorized into three sets: rectangle, cylindrical, and spherical (Figure 10). The experimentation begins by capturing data from sensors mounted on the data glove when grasping different objects. Each sensor’s data was recorded with a sample length of 10,000 ms. This means that for every sensor, a continuous stream of measurements was recorded for a duration of 10 s. The ADXL335 sensor data was collected at a frequency of 7 Hz. A total of 1550 samples were collected as raw data for each object. After recording the raw data from the sensors, we proceeded to split them into training and testing sets using an 80:20 ratio.

During the data analysis step, the sensor data was fused before being used as input for the machine learning (ML) model. The TinyML model was developed using a preprocessing feature extraction method in the development phase [18]. Following the training phase, the ML model was tested on the test dataset. To optimize the performance of the ML model, the number of training cycles was increased to 250, and the learning rate was set to 0.0005. Finally, the developed model was deployed on a real robotic hand apparatus for grasping objects of various shapes and sizes.

### 3.1. Data Collection 

The human demonstrator grabbed each item for 10 s. The accelerometer was utilized to acquire hand data in this study, with a frequency of 7 Hz. The sensors were positioned on the second segment of each finger. Figure 11 illustrates a few samples of the data acquired by the data glove for grasping different objects. As illustrated in Figure 11a, the thumb angle for grasping rectangle-shaped objects ranges from 18 to 21 degrees, while the angles for the ring and pinky fingers vary between 45 and 51 degrees. Similarly, for grasping spherical-shaped objects (Figure 11b), the thumb angle ranges from 15 to 17 degrees, and the angles for the ring and pinky fingers vary between 55 and 58 degrees. When it comes to grasping cylindrical-shaped objects, as shown in Figure 11c, the thumb, ring, and pinky finger angles vary between 20–22 degrees, 55–58 degrees, and 43–45 degrees, respectively. Notably, the index finger angle, as shown in Figure 11c, differs significantly from the angles observed when grasping rectangular- and spherical-shaped objects, ranging from 29 to 30 degrees.

### 3.2. Data Analysis and Feature Extraction 

In this paper, the spectral analysis is utilized for data analysis and feature extraction. Low-pass and high-pass filters are applied to filter out unwanted frequencies. Spectral analysis is a great tool for analyzing repetitive motion, such as data from accelerometers in our case. It extracts the features based on frequency and power characteristics of a signal over time. The wavelet transformation is often preferred over Fourier transformations in certain applications due to its advantages in examining specific frequencies and reducing computational requirements.

We used the following wavelet transform to extract the features.
(7)$$F\left(\tau ,s\right)=\frac{1}{\sqrt{|s|}}\int_{-\infty }^{+\infty }f\left(t\right)\psi^{*}\left(\frac{t-\tau }{s}\right)dt$$
where τ is scaling factor and *s* represents time shift factor. Figure 12 depicts the feature extraction from the accelerometer data using wavelet transform for a sample time of 5 s.

### 3.3. Neural Network Classifier

A neural network classifier takes some input data and outputs a probability score that indicates how likely it is that the input data belongs to a particular class. The neural network is composed of multiple layers, each containing numerous interconnected neurons. The network’s output is compared to the correct answers, and based on the results, the weights of the connections between neurons are modified. This iterative process is repeated until the network effectively learns to predict the correct answers for the training data [18]. Figure 13 shows the neural network architecture employed in this paper. We used a shallow feed-forward neural network (NN) composed of three layers. The input layer consists of 140 neurons, which are determined by the dimensions in the input data. The hidden layer is composed of layers, each having 10 neurons with hyperbolic tangent activation functions. The output layer has three neurons with the softmax transfer function, which corresponds to the number of classes (cylinder, rectangle, sphere). For training the NN, we use the gradient descend method and the Adam optimizer from the *tensorflow.keras.optimizers* library. A batch size of 32 is considered for the training process. Table 3 shows the neural network classifier setting parameters. The training was performed on a personal computer with an Intel^®^ Core™ i7 processor with 8 GB of RAM memory.

To evaluate our model performance, we used a confusion matrix, which tabulates a summary of the correct and incorrect responses generated by the model when presented with a dataset. The labels on the side of the matrix represent the actual labels present in each sample, while the labels on the top represent the predicted labels produced by the model. The confusion matrix is illustrated in Figure 14. The overall accuracy of the trained model is 95.7%, which is acceptable for our application. The spatial distribution of input features is shown in Figure 15. Green-color items are classified correctly, whereas items in red color are misclassified. Hence, the wavelet-based function proposed in Section 3.2 demonstrates its effectiveness in effectively distinguishing between the different classes.

By analyzing the results of grasping, it was observed that accelerometer sensors, which capture the dynamic movements and orientations of the grasped objects, can effectively differentiate between spheres and rectangles or cylinders. This is due to fact that the rolling motion of a sphere during a grasp generates distinctive accelerometer patterns compared to the linear movements associated with grasping rectangles or cylinders. Moreover, we noticed that grasps on spheres resulted in more evenly distributed pressure across the contact area, while grasps on rectangles and cylinders exhibit higher pressure around the edges or corners due to the need for more precise finger positioning. Additionally, we observed that flexi sensors placed on the finger joints could capture the degree of finger flexion during grasping. Experiments showed that grasps on rectangles require more pronounced flexion in the fingers to achieve a stable grip, while grasps on spheres involve more extended finger positions.

### 3.4. Model Testing

The trained model undergoes testing using the test dataset to validate its performance. Table 4 shows the prediction results for the test dataset. The overall accuracy achieved during testing is 96.22%. To further improve the model’s accuracy, the samples that were initially part of the test dataset can be moved back to the training dataset, and the model can be retrained with these augmented training data.

Metric parameters, such as accuracy, precision, and recall, play a crucial role in evaluating the performance of classification models in machine learning. Accuracy measures the overall correctness of the model’s predictions. Precision indicates the model’s accuracy when it predicts the target class, while recall measures the model’s ability to correctly identify all instances of the target class. Equations (8)–(10) represent the formulas for calculating accuracy, precision, and recall. These metrics provide valuable insights into the effectiveness and reliability of a classification model.
(8)$$Accuracy=\frac{True\;Positive+True\;Negative}{True\;Positive+True\;Negative+False\;Positive+False\;Negative}$$
(9)$$Precision=\frac{True\;Positive}{True\;Positive+False\;Positive}$$
(10)$$Recall=\frac{True\;Positive}{True\;Positive+False\;Negative}$$

Figure 16 illustrates the performance metrics of the proposed model, considering different training settings by varying the number of training cycles. In this comparison, the learning rate value was kept constant at 0.0005. It can be seen that when the number of training samples is 150, the overall accuracy of the model reaches 84%. However, if the number of training samples is increased to 500, the model exhibits near-perfect accuracy, indicating overfitting. To address this issue, we set the number of training samples to 300, resulting in improved overall accuracy, precision, and recall values.

### 3.5. Experimentation Setup

The proposed HRI system is validated using a real robot arm with an attached robotic hand, as depicted in Figure 8b. In this study, the object class type (sphere, cylinder, rectangle) is manually provided as input to the model, and the model predicts the corresponding movements (angle in degrees) of the robotic hand’s fingers, which are mapped into servo angles ranging from 0 to 90 degrees. The average bending angles of servo joints of robotic hand fingers for grasping the objects are shown in Table 5. Figure 17 showcases the results of the robotic grasping of objects with different shapes and sizes. 

The performance of the proposed data glove is compared to other state-of-the-art data glove prototypes in Table 6. The results indicate that data gloves utilizing optical fiber sensors and IMU sensors outperform in measuring joint angles. However, IMU sensors generally offer a lower cost compared to optical fiber sensors. In our proposed design, we incorporate five IMU sensors (accelerometers) along with flex sensors to monitor the proximal interphalangeal (PIP) joint angle for each finger. Additionally, an optional IMU sensor is included to track wrist roll motion. 

Although a wearable glove-based HRI offers promising possibilities, it comes with certain limitations and challenges. Integrating multiple sensors into a wearable glove requires accurate calibration. To this end, we calibrated all sensors to ensure accurate and synchronized sensor measurements across different modalities (e.g., accelerometers, force sensors, flex sensors). Also, sensor measurements can be affected by noise or occasional sensor failures, leading to unreliable data. In this paper, we employed the sensor fusion technique to combine information from multiple sensors, improving overall accuracy and robustness. Moreover, proper sensor positioning is essential to capture relevant data and enable accurate interpretation of human hand movements and grasp patterns. By conducting several experiments, we optimized the sensor placement and orientation for improved HRI performance. 

For future work, we will incorporate a camera into the HRI system to capture images of objects and automatically detect object classes, enhancing its ability to recognize and manipulate objects. Additionally, we will deploy the flex and pressure sensors to adjust position and force of gripping with an industrial gripper. As can be seen in Figure 18, universal robots need position, speed, and force parameters, which should be set before object grasping.

## 4. Conclusions

This paper presents a practical human–robot interaction system based on a wearable glove for object grasp application. We embedded three different types of sensors to capture grasp information from human demonstrators to imitate the posture and dynamics of the human hand. Calibration algorithms were implemented to avoid errors caused by the sensors. After analyzing the collected data, the positions of the accelerometer, pressure, and flexi sensors were determined in the wearable glove to capture grasp information of objects with different sizes and shapes. A three-layer neural network was trained to recognize grasp orientation and position, based on the multi-sensor fusion method. The experimental results showed that an industrial robot can successfully grasp sphere, rectangle, and cylinder objects with the proposed HRI based on learning from demonstration.

## Figures and Tables

**Figure 1 sensors-23-09780-f001:**
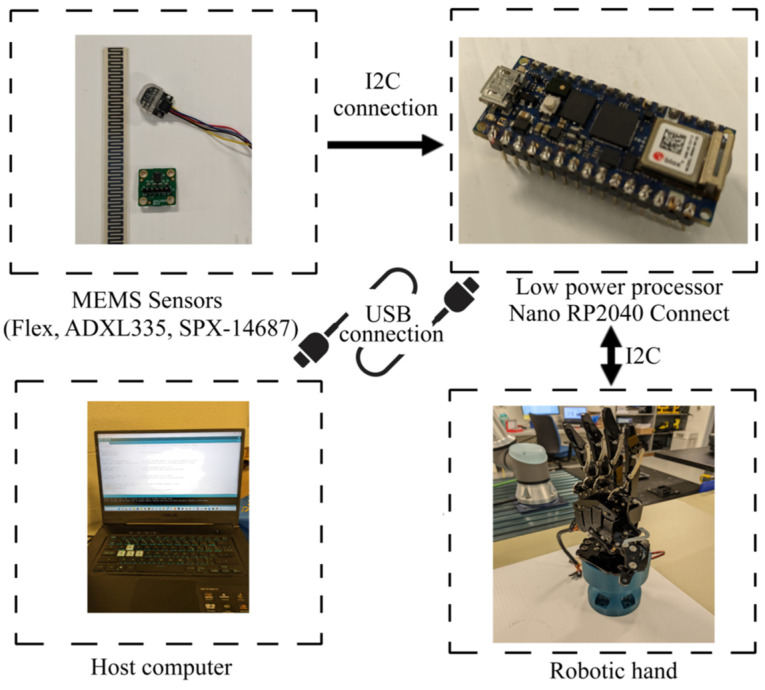
The communication flow of hardware system.

**Figure 2 sensors-23-09780-f002:**
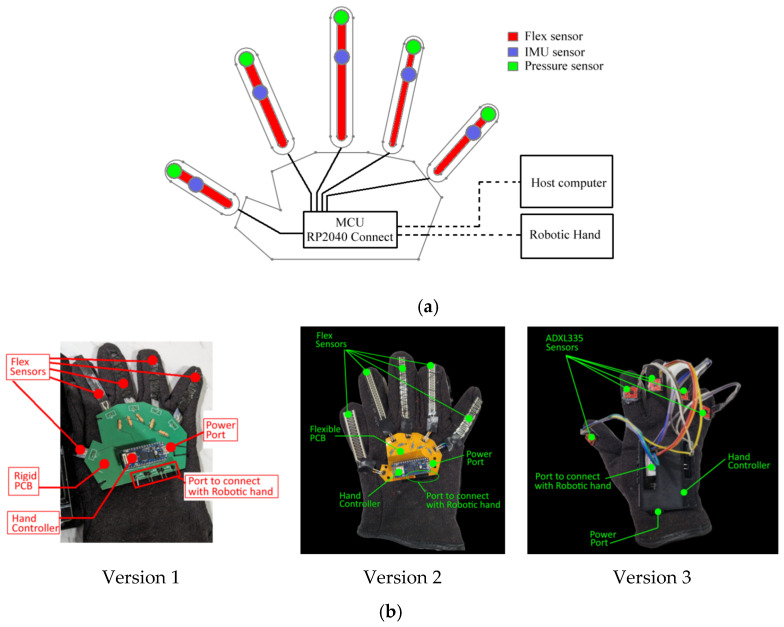
Structure of data glove; (**a**) schematic diagram of data glove. (**b**) different prototypes.

**Figure 3 sensors-23-09780-f003:**
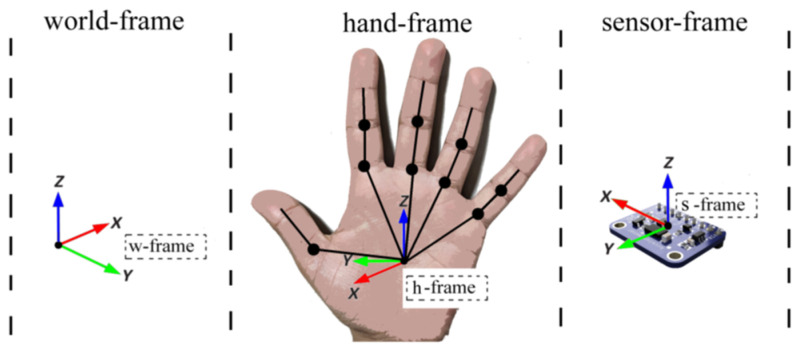
Definitions of the three frames.

**Figure 4 sensors-23-09780-f004:**
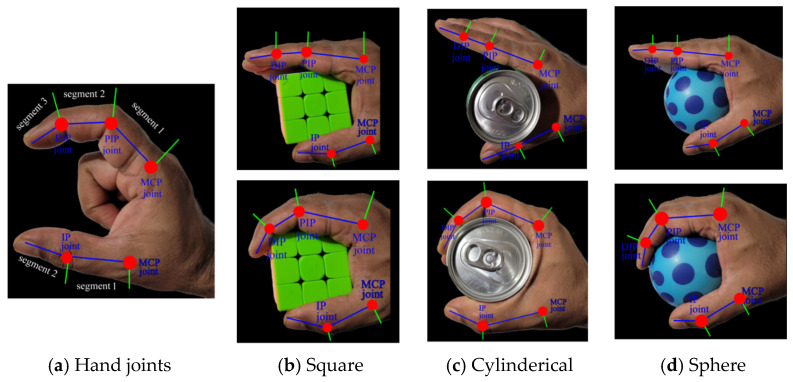
(**a**) Structure of hand joints, (**b**–**d**) grasping action for different objects.

**Figure 5 sensors-23-09780-f005:**
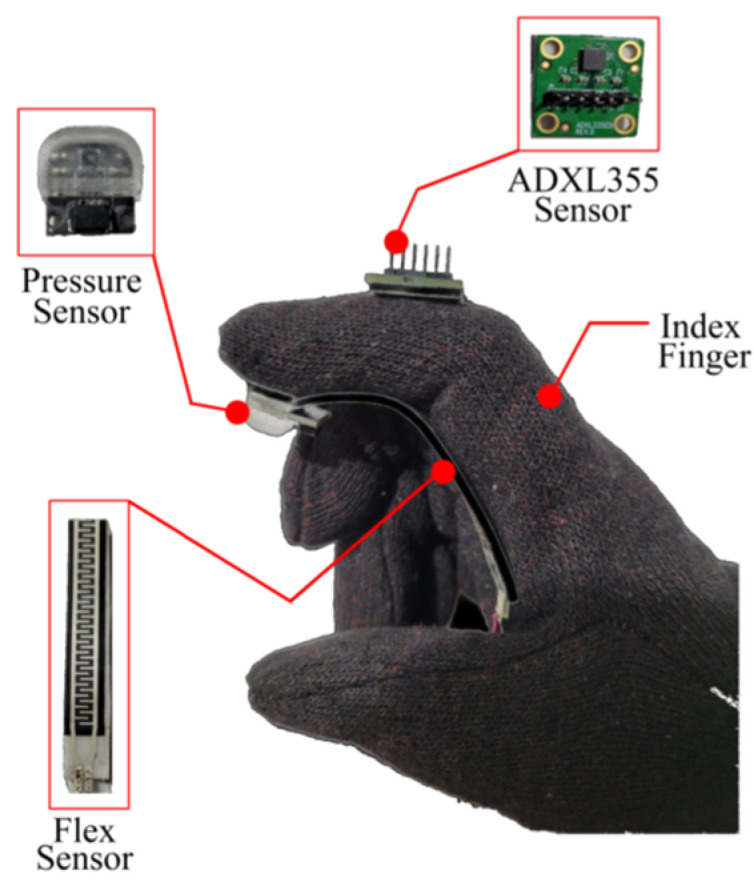
Sensor’s placement for index finger.

**Figure 6 sensors-23-09780-f006:**
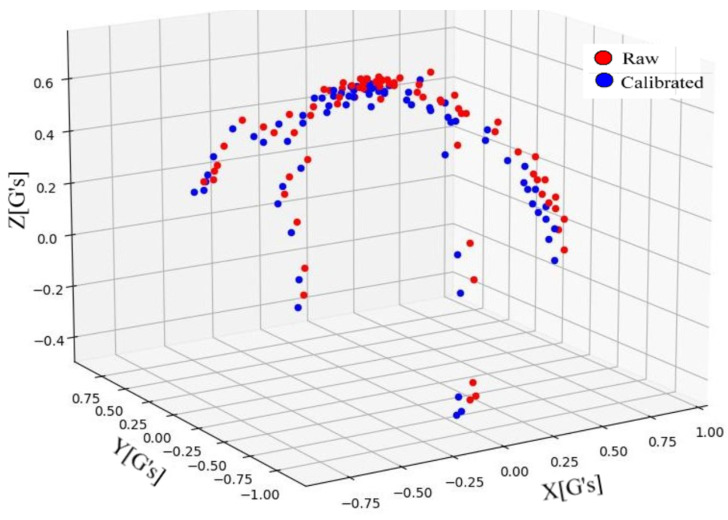
Comparisons of accelerometer data distribution before and after calibration.

**Figure 7 sensors-23-09780-f007:**
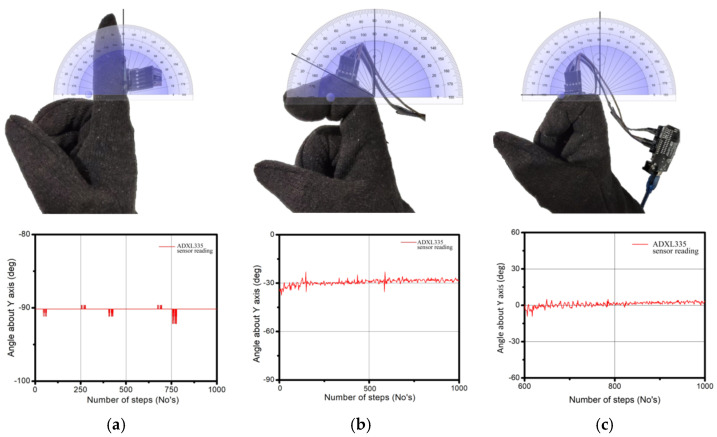
Comparisons of the measured angle between data glove accelerometer and drawing protector: (**a**) −90-degree, (**b**) −30 degree, and (**c**) 0 degrees.

**Figure 8 sensors-23-09780-f008:**
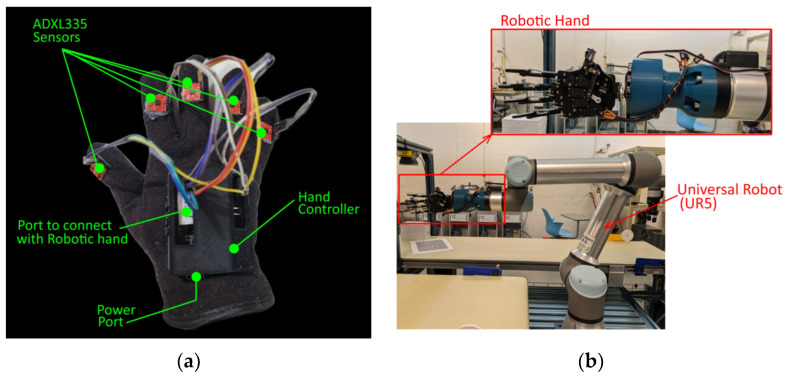
Human–robot interaction components: (**a**) data glove, and (**b**) robotic hand with UR5 robot.

**Figure 9 sensors-23-09780-f009:**
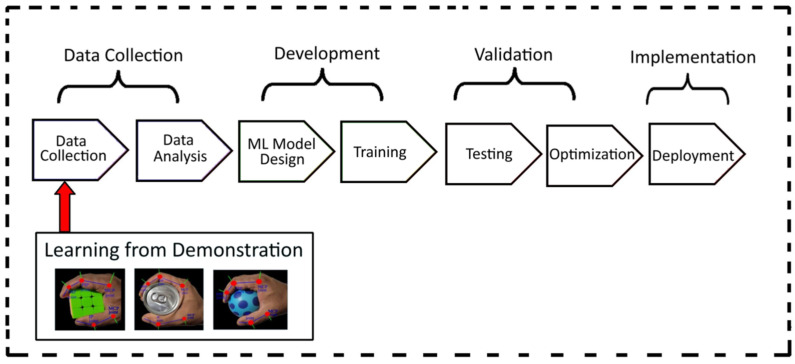
Flow diagram of ML model based on learning from demonstration.

**Figure 10 sensors-23-09780-f010:**
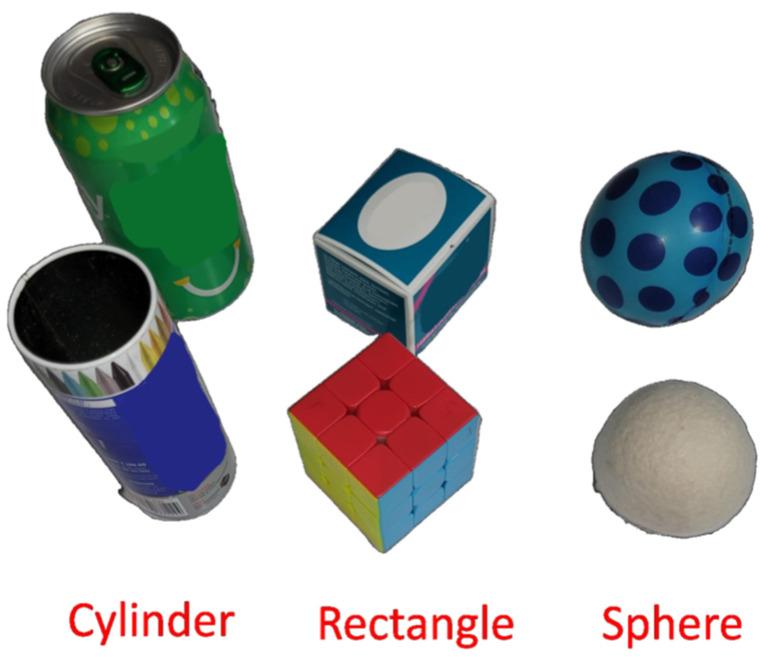
Everyday items for data capture.

**Figure 11 sensors-23-09780-f011:**
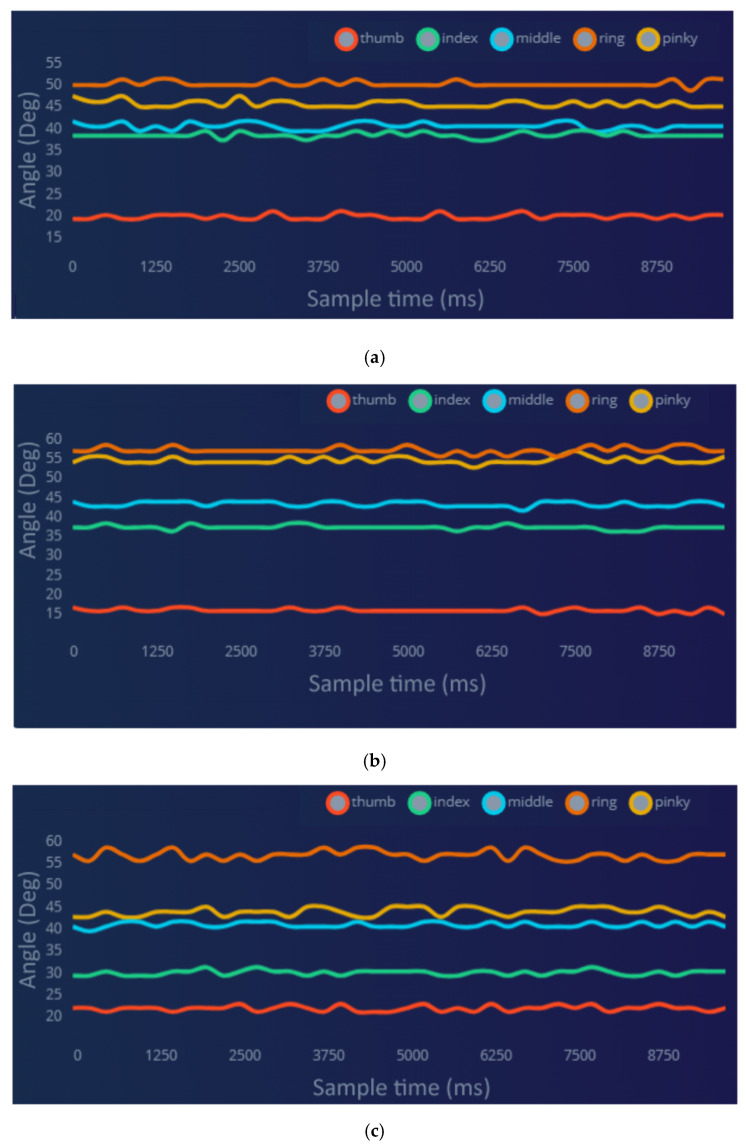
Raw data acquired by data glove: (**a**) rectangle objects, (**b**) spherical objects, and (**c**) cylindrical objects.

**Figure 12 sensors-23-09780-f012:**
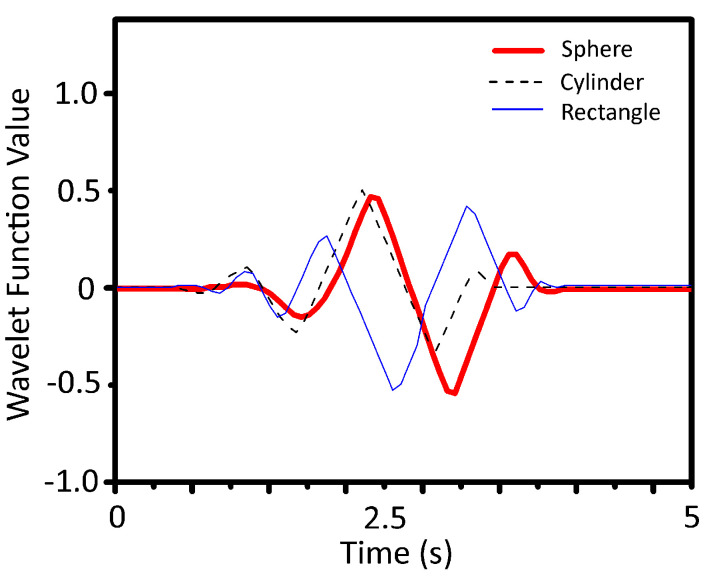
Feature extraction using wavelet transform.

**Figure 13 sensors-23-09780-f013:**
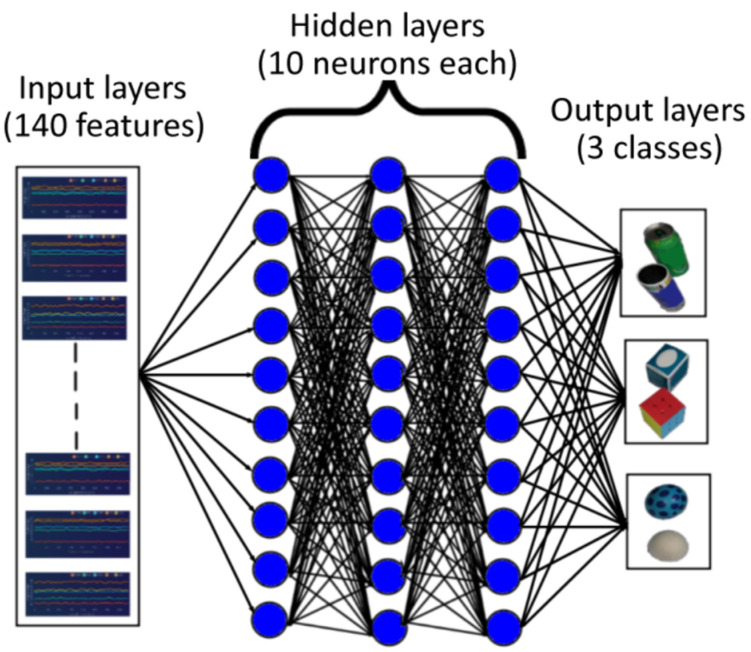
Neural network architecture.

**Figure 14 sensors-23-09780-f014:**
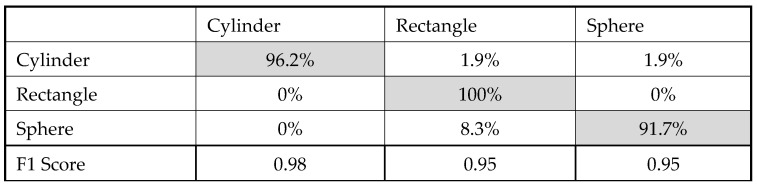
Confusion matrix.

**Figure 15 sensors-23-09780-f015:**
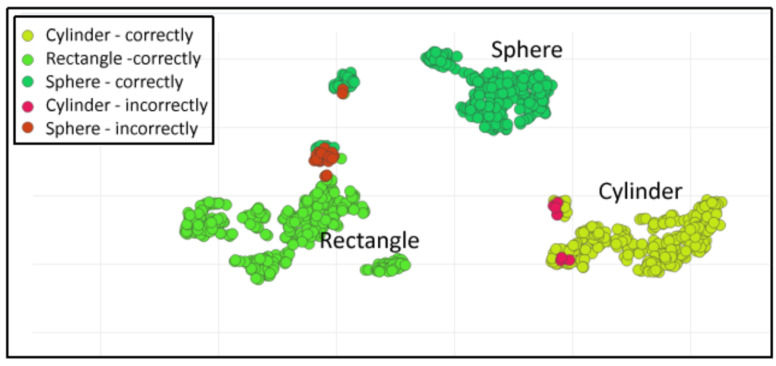
Spatial distribution of input features.

**Figure 16 sensors-23-09780-f016:**
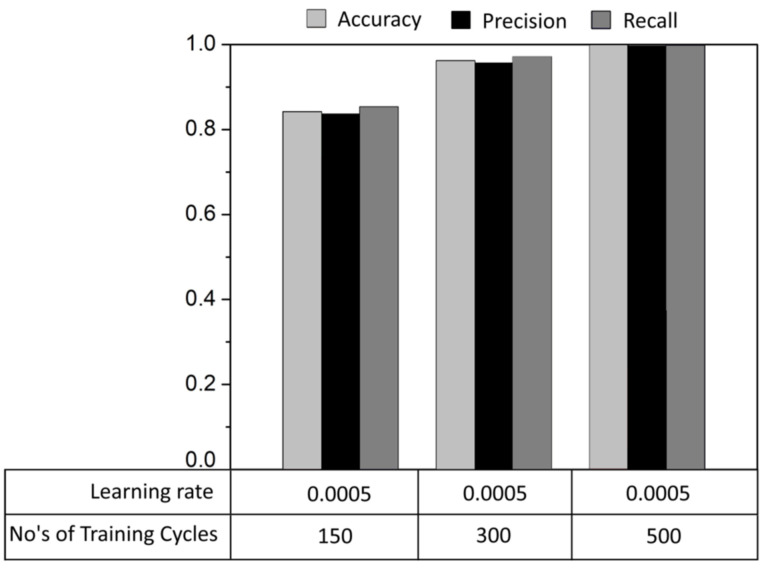
Performance matric.

**Figure 17 sensors-23-09780-f017:**
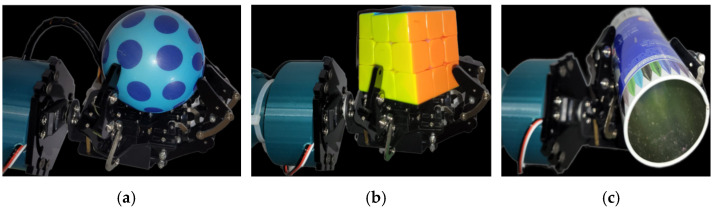
Grasping the objects with different shape and size: (**a**) sphere, (**b**) rectangle, and (**c**) cylinder.

**Figure 18 sensors-23-09780-f018:**
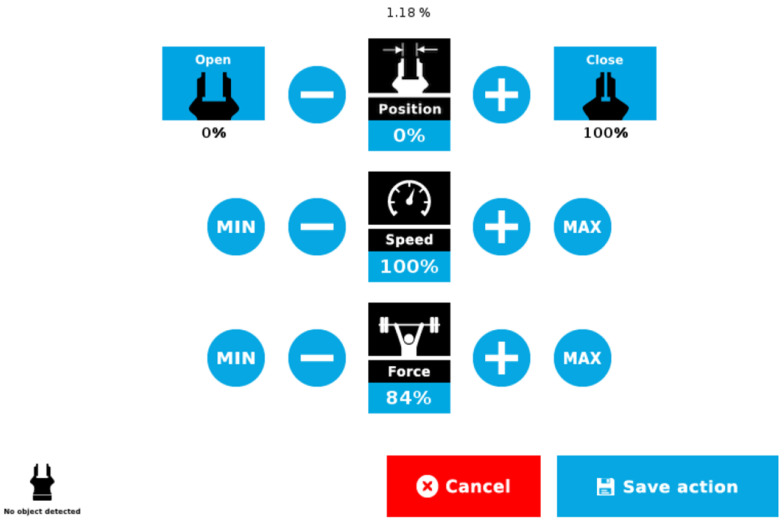
Universal robot interface for grasping.

**Table 1 sensors-23-09780-t001:** The specifications of sensors.

Units	ADXL335	Flex Seonsor	SPX-14687
Dimensions	3 axis	1 axis	1 axis
Power supply	1.8–3.6 V	1.8–3.6 V	2.5–3.6 V
Operating temperature	−40–+85 °C	−35–+80 °C	−40–+85 °C
Dynamic range	±3 g	-----------	-----------
Cross-axis sensitivity	±1%	-----------	-----------
Accuracy	±1%	±8%	±10%

**Table 2 sensors-23-09780-t002:** Measurement results of drawing protector and data glove accelerometer.

Angle from drawing protector (deg)	−90	−28.5
Average angle from data glove (deg)	−88.45	−28.12
Error (degree)	1.55	0.38
Error %	1.75%	1.35%

**Table 3 sensors-23-09780-t003:** Neural network classifier parameter values.

Parameters	Value
Number of training cycles	300
Learning rate	0.0005
Validation set size	20

**Table 4 sensors-23-09780-t004:** Test dataset validation.

Sample Name	Expected Lables	Time Length	Accuracy %	Results
cylinder.4dmd	cylinder	10 s	97%	36 cylinder, 1 other
rectangle.4dmf	rectangle	10 s	89%	33 cylinder, 4 others
sphere.4dme	sphere	10 s	100%	37 sphere
sphere.4dmt	sphere	10 s	94%	35 sphere, 2 others
cylinder.4dmdh	cylinder	10 s	100%	37 cylinder

**Table 5 sensors-23-09780-t005:** Bending angles of servo joints.

Objects Shape	Thumb	Index Finger	Middle Finger	Ring Finger	Pinky Finger
	(deg)	(deg)	(deg)	(deg)	(deg)
Sphere	14.2	36.46	34.42	40.13	42.12
Rectangle	18.59	39.62	42.93	55.70	52.81
Cylinder	22.45	32.43	44.52	57.20	45.19

**Table 6 sensors-23-09780-t006:** Comparison of proposed data glove.

Publications, Year	Type of Sensor	Number of Sensors	Joint Angle Deviation
Fei et al. [16], 2021	IMMU	12 (one hand)	1.4 deg
Cha et al. [19], 2017	Piezoelectric	19 (one hand)	5 deg
Li et al. [20], 2011	Optical encoder	14 (one hand)	1 deg
Proposed data glove	IMMU + flex	5 + 5 (one hand)	1.55 deg

## Data Availability

The data related to this paper are available on request from the corresponding author.

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
