# Peer review of "Human–Robot Interaction Using Learning from Demonstrations and a Wearable Glove with Multiple Sensors"

_sensors, 2023, doi:10.3390/s23249780_

Round 1

Reviewer 1 Report

Comments and Suggestions for Authors

This paper presents an interesting approach for human-robot interaction using a wearable glove with multiple sensors. The authors propose a sensor-based learning from demonstration method to enable a robotic hand to mimic human grasping of objects with different shapes. The key strengths of the paper are:

The paper clearly explains the motivation and objectives of the research, highlighting the importance of effective human-robot interaction for complex tasks like grasping.

The glove design incorporating accelerometer, pressure and flex sensors is novel and relevant for capturing motion and force data during grasping.

The sensor calibration process is described in sufficient detail. Experimental results validate the accuracy of the accelerometer measurements.

The placement of different sensors on the glove is optimized based on experiments with objects of different shapes and sizes. This is a nice practical approach.

The neural network classifier for learning grasp patterns shows reasonably good accuracy. Testing on a real robotic hand demonstrates the viability of the approach.

However, some aspects of the manuscript could be improved:

More details can be provided on the specific neural network architecture, optimizer used, training process etc. Only a high-level description is currently provided.

The results can be analyzed in more depth - for example, investigating which grasp patterns are more challenging to recognize. Confusion matrix provides some insights but more analysis would be useful.

Comparison with other state-of-the-art data gloves is useful but can be expanded. Metrics like accuracy, number of sensors, cost etc. can provide a more comprehensive comparison.

Reviewer 2 Report

Comments and Suggestions for Authors

The paper introduces a practical human-robot interaction (HRI) system that employs a wearable glove for object grasp applications. The system integrates three types of sensors – accelerometer, pressure, and flex sensors – to capture grasp information from human demonstrators, enabling the robot to imitate the posture and dynamics of human hands. The paper also addresses calibration algorithms to mitigate errors arising from sensor measurements. Additionally, a three-layer neural network is trained to recognize grasp orientation and position through a multi-sensor fusion method, and the experimental results claim successful grasping of various objects by an industrial robot.

One notable strength of the paper lies in its interdisciplinary approach, combining robotics, sensor technology, and machine learning to develop a comprehensive HRI system. The integration of multiple sensor types provides a rich source of data for the neural network, enhancing the system's ability to capture diverse grasping scenarios. The incorporation of calibration algorithms is a pragmatic step to mitigate sensor errors, emphasizing the paper's commitment to practical implementation.

However, the paper is not without its limitations and areas for improvement. Firstly, the paper lacks a thorough exploration of the limitations and challenges associated with the proposed system. Practical implementation of such systems may face issues related to real-world variability, diverse environmental conditions, and adaptability to unforeseen scenarios. A more robust discussion of these challenges and potential solutions would enhance the paper's credibility.

Furthermore, the paper could benefit from a more detailed analysis of the neural network's performance. Metrics such as accuracy, precision, and recall should be presented to provide a comprehensive understanding of the system's reliability and its ability to generalize to different objects and contexts. Additionally, the paper should discuss potential limitations of the neural network, such as overfitting or sensitivity to variations in input data.

The experimental results claim success in grasping various objects, but the paper lacks a comparison with existing methods or benchmarks. Including such comparisons would help assess the novelty and effectiveness of the proposed system in comparison to other approaches in the field.

In conclusion, the paper presents an intriguing and interdisciplinary approach to human-robot interaction using a wearable glove for object grasp applications. While the integration of multiple sensors and the use of a neural network for learning from demonstration are commendable, a more in-depth analysis of system limitations, thorough performance metrics, and comparative evaluations would strengthen the paper's contribution to the field.
